# DragSolver: A Multi-Scale Transformer for Real-World Automotive Drag Coefficient Estimation

**Ye Liu** [1 2]    **Yuntian Chen** [3 4]

## Abstract

Automotive drag coefficient ($C_d$) is pivotal to energy efficiency, fuel consumption, and aerodynamic performance. However, costly computational fluid dynamics (CFD) simulations and wind tunnel tests struggle to meet the rapid-iteration demands of automotive design. We present DragSolver, a Transformer-based framework for rapid and accurate $C_d$ estimation from large-scale, diverse 3D vehicle models. DragSolver tackles four key real-world challenges: **(1)** multi-scale feature extraction to capture both global shape and fine local geometry; **(2)** heterogeneous scale normalization to handle meshes with varying sizes and densities; **(3)** surface-guided gating to suppress internal structures irrelevant to external aerodynamics; and **(4)** epistemic uncertainty estimation via Monte Carlo dropout for risk-aware design. Extensive evaluations on three industrial-scale datasets (DrivAerNet, DrivAerNet++, and DrivaerML) show that DragSolver outperforms existing approaches in accuracy and generalization, achieving an average reduction of relative $L_2$ error by 58.7% across real-world datasets. Crucially, DragSolver is the first to achieve reliable, real-time $C_d$ inference on production-level automotive geometries.

## 1. Introduction

The aerodynamic drag coefficient ($C_d$) is a critical metric in automotive design, shaping energy efficiency, fuel consump-

---

[1]Shanghai Jiao Tong University, Shanghai 200240. PR China [2]Zhejiang Key Laboratory of Industrial Intelligence and Digital Twin, Eastern Institute of Technology, Ningbo, Zhejiang 315200, P.R. China [3]Ningbo Institute of Digital Twin, Eastern Institute of Technology, Ningbo, Zhejiang 315200, P.R. China [4]Ningbo Key Laboratory of Advanced Manufacturing Simulation, Eastern Institute of Technology, Ningbo, Zhejiang 315200, P.R. China. Correspondence to: Yuntian Chen <ychen@eitech.edu.cn>.

*Proceedings of the 42nd International Conference on Machine Learning*, Vancouver, Canada. PMLR 267, 2025. Copyright 2025 by the author(s).

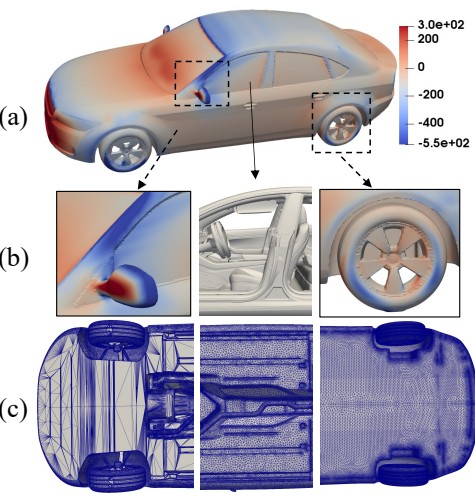

*Figure 1.* Key challenges in learning-based $C_d$ prediction. (a) Multi-scale influence: Global shape influences $C_d$ by determining large-scale aerodynamic flow patterns, including pressure distribution and wake formation. (b) Local and irrelevant structures: Small details (e.g., mirrors, wheels) induce turbulence, while the inset highlights internal components (e.g., seats, dashboard) that are irrelevant to external aerodynamics but appear in 3D scans. (c) Geometric and mesh variability: Differences in vehicle underbody structures, mesh resolution, and discretization highlight the challenge of inconsistent representations across datasets, necessitating robust normalization.

tion, and vehicle range (Sudin et al., 2014). In the early design stages, rapid and accurate $C_d$ estimation is essential for evaluating shape variants under tight development timelines. Traditional numerical methods, such as computational fluid dynamics (CFD) simulations and wind tunnel experiments, provide reliable estimations (Rodi, 1997; Baker & Brockie, 1991) but are computationally expensive, often requiring hours or even days per iteration for complex 3D models (Moin & Mahesh, 1998). This bottleneck significantly hinders the iterative workflows crucial to modern automotive engineering.

From a physics standpoint, the drag force $F_D$ on a vehicle is

$$F_D = \tfrac{1}{2}\,\rho\,v^2\,C_d\,A, \tag{1}$$

where $\rho$ is air density, $v$ is velocity, and $A$ is frontal area.

Under standardized test conditions, $\rho$ and $v$ remain fixed, leaving $C_d$ as the primary shape-dependent variable. If one could directly map a vehicle's 3D geometry to $C_d$, engineers could swiftly assess design alternatives without incurring the high costs of CFD or wind tunnel experiments.

Despite this appeal, developing a real-world practical geometry-to-$C_d$ predictor faces four critical challenges (Sudin et al., 2014): (1) Multi-scale interactions: Local details (e.g., spoilers, side mirrors) and global shape factors (e.g., roof curvature) both heavily influence drag. (2) Geometric Diversity: Vehicles differ widely in size, proportion, and mesh density, requiring robust normalization across heterogeneous data. (3) Irrelevant Internal Structures: Real-world test samples may include interior components (e.g. seats, steering wheel) absent in the training set of exterior surfaces, adding noise and hindering focus on flow-dominant geometry; (4) Predictive uncertainty: Industrial pipelines need confidence estimates for risk-aware design, but most existing ML methods treat $C_d$ prediction as a "black box." Figure 1 illustrates these issues in detail.

Conventional CFD and wind tunnel experiments, though accurate, remain prohibitively expensive for repeated design iterations (Moin & Mahesh, 1998; Heinz, 2020; He et al., 2021). Recent deep-learning surrogates (Rios et al., 2021; Abbas et al., 2022) can accelerate aerodynamic analysis but still exhibit key limitations: (1) existing $C_d$ prediction methods (Song et al., 2023; Elrefaie et al., 2024a) often rely on 2D views or low-dimensional shape parameters, failing to capture real-world 3D geometry; (2) Transformer-based 3D networks (Vaswani et al., 2017; Pang et al., 2022; Han et al., 2024; Chen et al., 2024; Wu et al., 2024b; 2022; Wang, 2023) excel at modeling global context via multi-head attention but typically assume uniform point densities and lack robust mesh normalization or interior filtering; and (3) neural PDE solvers (Li et al., 2021; 2023c; Hao et al., 2023; Wu et al., 2024a) emphasize local flow states rather than a global scalar like $C_d$, requiring cumbersome flow-field integration for large, unstructured car meshes. Thus, while Transformers inherently excel at capturing distant geometric cues (e.g., mirror-to-tail or underbody interactions) that strongly influence aerodynamics, a practical geometry-to-$C_d$ pipeline still needs dedicated modules to handle heterogeneous mesh scales, interior structures, and predictive uncertainty.

In response, we introduce DragSolver, a practical deep learning system that delivers reliable, real-time $C_d$ estimations for diverse, large-scale automotive models. Specifically, Specifically, DragSolver integrates: (1) Multi-scale feature extraction to capture both global shape contexts and local geometric cues crucial for aerodynamics. (2) Heterogeneous scale normalization to handle drastically varying mesh densities and vehicle sizes consistently. (3) Surface-guided gating to automatically suppress or ignore irrelevant

internal structures, focusing on exterior flow-dominant surfaces. (4) uncertainty estimation via a lightweight Monte Carlo dropout scheme, enabling risk-aware design decisions in industrial workflows. We extensively evaluate Drag-Solver on three real-world datasets (DrivAerNet, DrivAerNet++, and DrivaerML). Results show an average 58.7% relative $L_2$ improvement over existing baselines. Furthermore, DragSolver achieves near real-time inference ($\sim$0.9–5 s per shape) on a single GPU, significantly undercutting hours-long CFD pipelines. To foster broader research and real-world adoption, we will release our code upon acceptance. Overall, our contributions are threefold:

- We present the first end-to-end multi-scale transformer-based neural solution for automotive drag forecasting on large, unstructured industrial datasets.

- We design a surface-guided gating mechanism with heterogeneous normalization and uncertainty estimation, addressing real-world geometric complexities.

- We validate on three high-fidelity, diverse datasets, consistently surpassing prior methods in accuracy, speed, and out-of-distribution generalization.

## 2. Related Work

**Automotive Aerodynamic Analysis.** Accurate drag coefficient ($C_d$) estimation traditionally relies on *computational fluid dynamics* (CFD) and *wind tunnel* testing (Menter et al., 2003; Scardovelli & Zaleski, 1999; Fröhlich & Von Terzi, 2008). Though these methods yield high-fidelity aerodynamic data, they are slow and costly, limiting rapid design iteration. While recent learning-based approaches (Song et al., 2023; Elrefaie et al., 2024a;b) reduce CFD costs, most rely on low-dimensional shape parameters (e.g., a few geometric descriptors) or 2D projections (e.g., silhouette views). Such simplifications ignore the intricate 3D structure that significantly impacts aerodynamic flow. Consequently, these methods struggle to capture local flow phenomena, limiting their applicability in real-world design loops.

**Neural PDE Solvers.** A related direction involves neural PDE solvers (Karniadakis et al., 2021; Wang et al., 2023), serving as fast surrogates for PDE-governed phenomena such as fluid flow or stress fields. Neural operators methods (Li et al., 2021; 2023d; Wu et al., 2023; Li et al., 2023b) and Transformer-based PDE solvers (Liu et al., 2022; Li et al., 2023c; Hao et al., 2023; Wu et al., 2024a) achieve strong results in predicting local flow states. However, deriving a global scalar like $C_d$ typically requires further integration of the entire flow field or explicit knowledge of pressure/velocity distributions, making them less direct and often inaccurate for global drag prediction. In contrast, DragSolver bypasses explicit PDE modeling by learning a

geometry-to-$C_d$ mapping end-to-end, focusing on capturing the global shape cues relevant to drag without solving local fields and better suits large-scale automotive meshes with partial interior data.

**Geometric Deep Learning**  Deep learning on 3D data has progressed through point-based, voxel-based, mesh-based, and more recently *transformer-based* paradigms. **Point-based** networks (Qi et al., 2017a;b; Zhao et al., 2021; Pang et al., 2022; Han et al., 2024; Chen et al., 2024) directly process unstructured point clouds but can struggle with very dense or uneven distributions. Voxel-based methods transform point clouds into regular voxel grids to facilitate 3D convolution operations (Wu et al., 2024b; 2022; Wang, 2023). Mesh-based approaches (Pfaff et al., 2021b; Wang et al., 2019) exploit edge convolutions, though they can be sensitive to mesh topology changes and not trivially adopt multi-scale feature fusion. However, none of these mainstream methods directly address (1) heterogeneous mesh normalization, (2) gating out irrelevant internal geometry, or (3) automotive-specific $C_d$ regression with uncertainty. Our proposed DragSolver extends the transformer-based pipeline, integrating multi-scale hierarchical encoding (Wu et al., 2024b) and additional modules tailored to real-world automotive drag estimation.

**Predictive Uncertainty Estimation**  Uncertainty-aware ML helps practitioners gauge the reliability of estimations. Bayesian neural networks (Jospin et al., 2022) are a classic approach, but often complicated to implement and scale. MC Dropout (Gal & Ghahramani, 2016) offers a simpler variational approximation: applying dropout at inference time for multiple forward passes. Ensemble methods (Lakshminarayanan et al., 2017) also provide robust uncertainty but at higher memory cost. In our setting, epistemic uncertainty guides risk-aware decisions when deploying $C_d$ estimations in industrial design pipelines.

## 3. Methodology

We present DragSolver, a novel Transformer-based approach for real-time drag coefficient estimation from large-scale 3D automotive geometries. Our design addresses four key challenges in aerodynamic optimization: multi-scale interactions, heterogeneous geometry, irrelevant internal structures, and predictive uncertainty.

### 3.1. Problem Formulation
**Task Definition.**  We aim to learn a function $f(\mathbf{G}) \rightarrow C_d$ mapping an automotive geometry $\mathbf{G}$ to its drag coefficient $C_d$. Concretely, each sample $(\mathbf{G}, C_d)$ provides: (1) a 3D representation of the vehicle (e.g., triangular mesh, point cloud, or voxel grid), and (2) a scalar $C_d$ measured or computed under standardized conditions (e.g., fixed air density $\rho$

and velocity $v$, cf. Eq. 1). Our goal is to approximate $C_d$ accurately and efficiently, avoiding the high cost of full-fledged CFD or wind tunnel experiments.

**Input Geometry.**  Let $\mathbf{G} = \{\mathbf{v}_i\}_{i=1}^N$ denote either $N$ surface vertices or an $N$-point cloud of the vehicle. In practice, $\mathbf{G}$ may include partial *interior* geometry (e.g. seats, steering wheel) that does not affect external aerodynamics. We address this via *surface-guided gating* (Sec. 3.4) to deemphasize irrelevant internal points, focusing on the flow-dominant exterior.

**Learning Objective.**  Denote by $\hat{C}_d = f_\theta(\mathbf{G})$ the predicted drag coefficient from a network with parameters $\theta$. We adopt a standard regression loss:

$$\mathcal{L}_{\text{reg}} = \frac{1}{M} \sum_{j=1}^M (\hat{C}_d^{(j)} - C_d^{(j)})^2, \qquad (2)$$

where $M$ is the mini-batch size. Additional terms for gating or regularization may be included (Sec. 3.4). Our goal is to make $\hat{C}_d$ closely match the true $C_d$ while remaining robust to geometric variations and interior noise.

### 3.2. Multi-Scale Architecture
**Motivation.**  Automotive drag depends on both *global* shape (e.g., roof curvature, body length) and *local* details (e.g., spoilers, side mirrors). A single-scale network may miss fine geometry or lose broad context. Hence, we adopt a multi-stage encoder design, inspired by Transformer-based point processing (Wu et al., 2024b), capturing both large and small-scale patterns.

**Stage-Wise Forward.**  As illustrated in Fig. 2, we begin with an embedding block that transforms raw points/voxels into an initial feature space. We then stack $L$ encoding stages $\{\text{enc}[0], \ldots, \text{enc}[L-1]\}$, each combining sparse or point-based convolutions with attention layers. Let $\mathbf{F}^{(0)}$ be the output of the embedding block. For $k = 1, \ldots, L$:

$$\mathbf{F}^{(k)} = \text{enc}[k]\big(\mathbf{F}^{(k-1)}\big). \qquad (3)$$

Deeper stages expand receptive fields to capture global shape context, while earlier features preserve local detail.

**Global Pooling at Each Stage.**  Unlike single-resolution pipelines, we collect multi-scale global features by applying a global pooling operator (GPool) at each stage output:

$$\mathbf{g}^{(k)} = \text{GPool}\big(\mathbf{F}^{(k)}\big), \quad k = 0, \ldots, L. \qquad (4)$$

Where $\mathbf{g}^{(k)} \in \mathbb{R}^{C_k}$ encodes the entire shape at scale $k$. Concatenating these pooled features

$$\mathbf{G}_{\text{multi-scale}} = \big[\mathbf{g}^{(0)} \parallel \mathbf{g}^{(1)} \parallel \ldots \parallel \mathbf{g}^{(L)}\big] \qquad (5)$$

yields a comprehensive representation capturing both shallow local cues and deep global context. We feed $\mathbf{G}_{\text{multi-scale}}$ into a final regression MLP to predict $\hat{C}_d$.

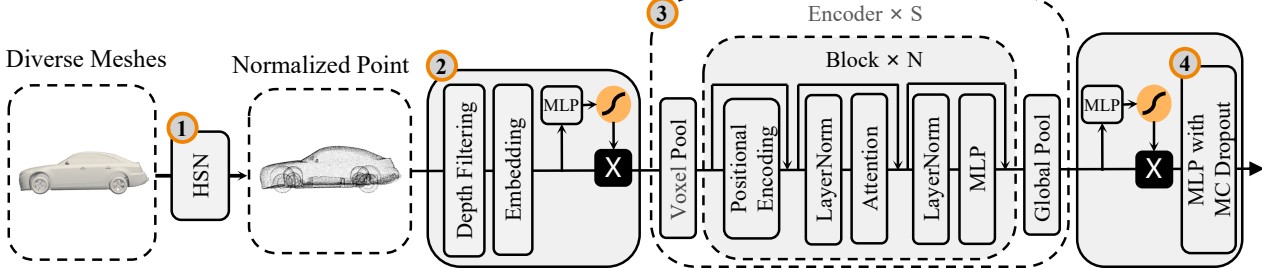

*Figure 2.* **Overview of DragSolver.** The framework consists of four main components: (1) **Heterogeneous Scale Normalization (HSN)**, which processes diverse vehicle meshes into a unified representation (Sec. 3.3). (2) **Surface-Guided Gating**, which filters out irrelevant internal structures to ensure a focus on aerodynamic surfaces (Sec. 3.4). (3) **Multi-Scale Architecture**, an encoder leveraging voxel pooling, positional encoding, and attention to capture both global and local aerodynamic features (Sec. 3.2). (4) **Epistemic Uncertainty Estimation**, utilizing Monte Carlo Dropout (MC Dropout) to provide confidence estimates in $C_d$ predictions (Sec. 3.5).

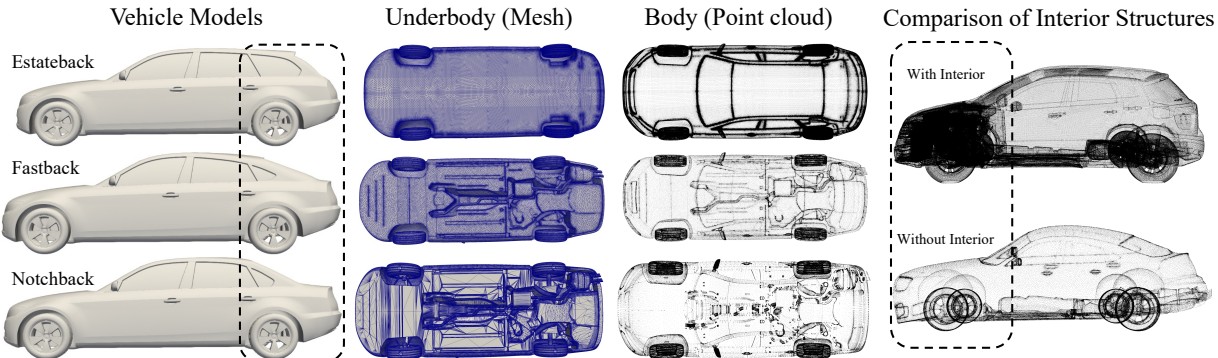

*Figure 3.* **Variability in vehicle configurations, mesh resolution, and structural representations.** The first column illustrates three distinct vehicle types: **Estateback** (top), **Fastback** (middle), and **Notchback** (bottom), highlighting variations in shape distribution. The second column presents **underbody meshes**, revealing significant differences in structural complexity and mesh resolution across datasets. The third column visualizes **point cloud representations**, showing heterogeneous surface discretization levels. The fourth column compares vehicles **with** and **without interior structures**, emphasizing distributional shifts where real-world data includes internal components irrelevant to external aerodynamics. These differences underscore the challenges posed by dataset heterogeneity in learning-based $C_d$ prediction.

## 3.3. Heterogeneous Scale Normalization

**Challenge.** Real-world vehicles differ drastically in physical scale (e.g., compact cars vs. large SUVs) and mesh densities (hundreds of thousands up to millions of polygons), causing inconsistent feature distributions if ingested naively.

**Automotive Geometric Normalization.** We employ a two-stage approach: (1) Wheelbase Alignment & Global Scaling. Since wheelbase is relatively standardized, we first rescale each shape so that the wheelbase matches a fixed length, ensuring consistent front/rear alignment. We then normalize all data by scaling the bounding box, ensuring consistency and reducing large-scale discrepancies. (2) Hybrid Resolution Sampling. Even after global scaling, some vehicles remain overly dense, others sparse. We mix mesh-based sampling (capturing high-curvature detail) with uniform sampling (mitigating density imbalance). This yields robust coverage of local geometry while preventing

over-/under-representation. Consequently, different vehicles become more comparable, aiding stable multi-scale feature extraction.

## 3.4. Surface-Guided Gating

**Motivation.** In practice, real-world scans or CAD files can include internal components (e.g., seats, steering wheel) that do not affect external aerodynamic flow. Training datasets often provide only exterior surfaces, whereas test models might unexpectedly contain such interior geometry, causing noise and performance degradation. Manual preprocessing is generally impractical for large-scale automotive designs. Therefore, we combine a *depth-based filtering* step with a *multi-stage gating* mechanism to robustly suppress irrelevant internal structures.

**Depth-Based Filtering (Preprocessing).** First, we perform a coarse elimination of clearly internal voxels. We estimate the depth of each voxel relative to the outer sur-

face using a ray-marching approach along the principal axes (e.g., $x$-direction). Specifically, we cast rays outward from a reference plane and track the sequence of voxel crossings. For each voxel, its depth is defined as the number of intersections encountered before reaching the outer shell. Voxels that are clearly distant from the surface—beyond a predefined threshold—are classified as internal and removed. This method efficiently eliminates large chunks of internal geometry while maintaining conservativeness to avoid mistakenly discarding subtle external cavities or enclosed aerodynamic features.

**Multi-Stage Gating Mechanism.** After filtering out obvious internal voxels, we assign an *initial gating weight* $c_i = 1$ to each remaining voxel or point. An MLP then refines $c_i$ at select stages of our multi-scale encoder. Specifically: (1) High-resolution input layer: Each voxel $i$ has $c_i^{(\text{init})} = 1$, which is updated by a gating MLP to account for uncertain regions near the exterior. Hence $c_i^{(\text{init})} \leftarrow \sigma(\text{MLP}(\mathbf{f}_i))$, letting the network softly downweight borderline internal geometry. (2) Mid-resolution layers: No gating is applied, allowing the model to freely attend to intermediate features without further constraints. (3) Final gating: Right before global pooling, we recompute $c_i^{(\text{final})}$ from the final-layer features. This second pass ensures a fine-grained re-check so that truly flow-relevant surface voxels remain prominent, while any residual internal points are minimized. If $c_i^{(l)} \approx 0$, that voxel's feature $\mathbf{f}_i^{(l)}$ is effectively nulled for subsequent aggregation. In the end, we compute

$$\mathbf{F}_{\text{global}} = \frac{\sum_i c_i^{(\text{final})} \mathbf{f}_i^{(\text{final})}}{\sum_i c_i^{(\text{final})} + \epsilon}, \tag{6}$$

focusing attention on truly external, aerodynamically relevant regions. During training, we insert random "ghost" voxels inside the bounding box to simulate unexpected interior structures.

### 3.5. Epistemic Uncertainty Estimation

**Rationale.** In industrial design loops, a single $\hat{C}_d$ estimation may be insufficient for critical decisions. Epistemic uncertainty indicates model ignorance, helping engineers weigh risk in choosing final shapes.

**MC Dropout.** We adopt Monte Carlo Dropout (Gal & Ghahramani, 2016), activating dropout at inference. Let $f_{\theta_k}(\mathbf{G})$ be the $k$-th forward pass with distinct dropout masks. Then

$$\overline{C_d} = \frac{1}{K} \sum_{k=1}^{K} f_{\theta_k}(\mathbf{G}), \quad \sigma_{C_d}^2 = \frac{1}{K} \sum_{k=1}^{K} \left(f_{\theta_k}(\mathbf{G}) - \overline{C_d}\right)^2. \tag{7}$$

Hence, $\sigma_{C_d}^2$ approximates epistemic uncertainty. In practice, $K = 10$ yields stable estimates with minimal overhead.

*Table 1.* Summary of the datasets used for evaluation.

| Dataset | Categories | #Car | #Mesh |
|---|---|---|---|
| DrivAerNet++ | Estateback Fastback Notchback | 7673 | 420k–2.2M |
| DrivAerNet | Fastback | 3760 | 420k |
| DrivaerML | Notchback | 483 | 750k |

## 4. Experiments

We conduct extensive experiments to evaluate DragSolver under both in-distribution (ID) and out-of-distribution (OOD) settings, reflecting real-world automotive design scenarios. Our primary goals are to (1) assess accuracy on known distributions, (2) measure generalization to unseen shapes, datasets, or interior structures, and (3) validate the effectiveness of each core module via ablation.

**Datasets.** We use three real-world datasets: (DrivAerNet++, DrivAerNet, and DrivaerML)covering diverse body shapes (estateback, fastback, notchback), mesh resolutions (420k to 2.2M polygons), and potential interior geometry. Table 1 summarizes key statistics. Each dataset reserves 20% for testing. For cross-dataset or cross-shape evaluations, we carefully partition training sets in one subset while testing on different subsets or entirely different datasets. More detailed descriptions of the datasets can be found in Appendix B.

**Evaluation Metrics.** For accuracy, we report: (1) Relative Ł$_2$ Error: highlighting percentage error relative to the ground-truth drag coefficient. (2) R$^2$ Score: capturing how much variance in $C_d$ is explained by the model,

$$R^2 = 1 - \frac{\sum_{i=1}^{M} \left(C_{d,i} - \hat{C}_{d,i}\right)^2}{\sum_{i=1}^{M} \left(C_{d,i} - \overline{C}_d\right)^2}, \tag{8}$$

where $\overline{C}_d$ is the mean of $\{C_{d,i}\}$ over the test set. A perfect model yields $R^2 = 1$, while $R^2 \leq 0$ indicates no explanatory power relative to the mean.

**Baselines.** We compare DragSolver with more than 10 baselines: including typical 3D geometric deep learning methods: PointNet (Qi et al., 2017a), PointNet++ (Qi et al., 2017b), PointTransformer (Zhao et al., 2021), Point-GPT (Chen et al., 2024), Mamba3D (Han et al., 2024), Point-Transformer V3 (Wu et al., 2024b), MeshGraphNet (Pfaff et al., 2021a), RegDGCNN (Elrefaie et al., 2024b). Including state-of-the-art neural operators: GNOT (Hao et al., 2023), GINO (Li et al., 2023a), Transolver (Wu et al., 2024a)

**Implementations.** For fairness, we implement a *five-stage* multi-scale encoder, each stage halving the spatial resolution (stride = 2) to progressively expand the receptive field.

*Table 2.* Performance comparison for in-distribution evaluation on two real-world datasets, where we record Relative $L_2$ and $R^2$. A smaller Relative $L_2$ is better, while a larger $R^2$ is better. For clarity, the best result is in **bold**, and the second-best is underlined. "Promotion" refers to the relative error reduction w.r.t. the second-best model, i.e. $1 - \frac{\text{Our error}}{\text{The next best error}}$. **Note**: DragSolver applies MC Dropout for uncertainty (see §3.5), yielding mean $\pm$ std. for each metric. Other baselines do not estimate model uncertainty and therefore report single values.

| Model | DrivAerNet | | DrivAerNet++ | |
|---|---|---|---|---|
| | Relative $L_2 \downarrow$ | $R^2 \uparrow$ | Relative $L_2 \downarrow$ | $R^2 \uparrow$ |
| PointNet (Qi et al., 2017a) | 0.0062 | 0.6184 | 0.0079 | 0.5170 |
| PointNet++ (Qi et al., 2017b) | 0.0046 | 0.7538 | 0.0064 | 0.6241 |
| PointTransformer (Zhao et al., 2021) | 0.0024 | 0.8844 | 0.0031 | 0.8015 |
| PointGPT (Chen et al., 2024) | 0.0023 | 0.8910 | 0.0028 | 0.8210 |
| Mamba3D (Han et al., 2024) | 0.0015 | 0.9173 | 0.0014 | 0.9192 |
| PointTransformer V3 (Wu et al., 2024b) | 0.0021 | 0.9051 | 0.0022 | 0.8901 |
| MeshGraphNet (Pfaff et al., 2021a) | 0.0230 | -0.2334 | 0.0238 | -0.5858 |
| RegDGCNN (Elrefaie et al., 2024b) | 0.0030 | 0.8418 | 0.0059 | 0.7064 |
| GINO (Li et al., 2023a) | 0.0286 | -0.4361 | 0.0316 | -0.5463 |
| Transolver (Wu et al., 2024a) | 0.0180 | -0.1646 | 0.0223 | -0.3931 |
| GNOT (Hao et al., 2023) | 0.0036 | 0.7926 | 0.0028 | 0.8312 |
| **DragSolver (Ours)** | **0.0007± 0.0001** | **0.9582 ± 0.0016** | **0.0005± 0.0001** | **0.9712 ± 0.0011** |
| Relative Promotion | 53.3% | 4.2% | 64.2% | 7.05% |

Channel dimensions grow from 32 to 256, and we employ robust patch-based attention, which ensures that our model parameter is comparable to other Transformer-based models such as Point Transformer V3 (Wu et al., 2024b). This hierarchical design balances local and global aerodynamic cues while efficiently handling diverse vehicle meshes. For Heterogeneous Scale Normalization (Sec. 3.3), the wheelbase is fixed at $2.8\,\mathrm{m}$, and we apply a hybrid sampling strategy that yields around $600\,\mathrm{k}$ points per vehicle. We enable Monte Carlo Dropout at the final regression layers with dropout rate $p = 0.1$. At inference, we draw $K = 10$ forward passes, yielding a set of predicted $C_d$ values $\{\hat{C}_d^{(k)}\}_{k=1}^K$. We then report *mean* $(\overline{C}_d)$ and *standard deviation* $(\pm\sigma_{C_d})$. We optimize with AdamW (Loshchilov, 2019) at a base learning rate of $1 \times 10^{-3}$, batch size 24, and train for 50 epochs, adopting cosine LR decay. All experiments run on NVIDIA A800 GPUs unless stated otherwise. Unless otherwise stated, these hyperparameters remain fixed across all experiments. See Appendix C for comprehensive descriptions of implementations.

### 4.1. Results of In-Distribution

**In-Distribution Performance.** We first evaluate each model in a same-distribution scenario, where training and test sets share the same shape/mesh statistics. Specifically, we independently train and test on DrivAerNet and DrivAerNet++ (Tab. 2). Point-based networks generally benefit from learning unstructured data but still suffer with fine geometry. PointGPT and Mamba3D attain higher $R^2$ on

DrivAerNet++, yet remain behind DragSolver. PDE-based or neural operator methods often focus on local flow states and require additional integration for a global $C_d$, leading to inconsistent or even negative $R^2$ on complex real-world shapes. Our DragSolver substantially outperforms all baselines on both datasets, reducing relative $L_2$ error to 0.0007 (DrivAerNet) and 0.0005 (DrivAerNet++). It achieves $R^2$ up to 0.9712, indicating an excellent fit with ground-truth labels. Compared to the second-best method, DragSolver yields a 53.3% and 64.2% promotion in error reduction, respectively, demonstrating its ability to consistently capture global aerodynamic interactions in purely data-driven manner. Overall, these in-distribution results confirm that our multi-scale network, aided by robust normalization excels at learning complex geometry-to-$C_d$ mappings.

### 4.2. Results of Out-of-Distribution

Although ID accuracy is important, practical deployment often faces unseen shape configurations or distinct mesh distributions. We thus evaluate Out-of-Distribution (OOD) settings where training and testing data differ in shape type, dataset origin, or internal structures. Figure 3 illustrates the diversity in real-world datasets, encompassing variations in vehicle shape, underbody complexity, and point cloud resolution. The presence of both high-density and low-density mesh structures, along with the inclusion of real-world interior components in some cases, creates significant distribution shifts. These differences pose challenges for learning-based prediction.

*Table 3.* **Cross-shape generalization** within DrivAerNet++. Train columns list body types used for training, Test indicates the held-out shape (estateback **E**, fastback **F**, notchback **N**).

| Model | Train | Test | Relative $L_2 \downarrow$ | $R^2 \uparrow$ |
|---|---|---|---|---|
| Mamba3D | F+N | E | 0.0097 | 0.4912 |
| PointGPT | F+N | E | 0.0117 | 0.3616 |
| RegDGCNN | F+N | E | 0.0124 | 0.3245 |
| **DragSolver** | F+N | E | **0.0031± 0.0003** | **0.8054± 0.0021** |
| Mamba3D | F+E | N | 0.0054 | 0.6179 |
| PointGPT | F+E | N | 0.0094 | 0.4984 |
| RegDGCNN | F+E | N | 0.0120 | 0.3838 |
| **DragSolver** | F+E | N | **0.0034± 0.0003** | **0.7842± 0.0019** |

*Table 4.* Performance comparison for **cross-dataset generalization**. DN++: DrivAerNet++, DML: DrivaerML. Models are trained on DN++ and tested on DML to evaluate generalization to unseen high-resolution meshes.

| Model | Train | Test | Relative $L_2 \downarrow$ | $R^2 \uparrow$ |
|---|---|---|---|---|
| Mamba3D | DN++ | DML | 0.0197 | -0.0684 |
| PointGPT | DN++ | DML | 0.0447 | -0.5447 |
| RegDGCNN | DN++ | DML | 0.0561 | -0.6145 |
| **DragSolver** | DN++ | DML | **0.0091± 0.0001** | **0.2732± 0.0004** |

**Cross-Shape within DrivAerNet++.** The DrivAerNet++ series includes Estateback, Fastback, and Notchback designs. Table 3 shows results where we train only on two body types (F, N) and test on the third (E)—absent from training, and vice versa. When testing on estateback (E) after training on fastback & notchback (F+N), DragSolver achieves the lowest relative $L_2$ (0.0031) and highest $R^2$ (0.8054), outperforming Mamba3D by an over 3× reduction in error. Similarly, for the F+E→N split, DragSolver again surpasses competing methods with an error of 0.0034 and $R^2 = 0.7842$. The substantial gap underscores the model's strong capacity to handle shape distribution shifts and capture exterior geometry cues that impact $C_d$.

**Cross-Dataset Generalization Results.** We next examine a *cross-dataset* scenario by training on DrivAerNet++ (DN++) and testing on DrivaerML (DML), where vehicles differ more substantially in shape design, mesh resolution, and potential internal complexities. Table 4 shows that DragSolver attains a much lower relative $L_2$ error (0.0091) and a positive $R^2$ of 0.2732, indicating it effectively transfers across dataset discrepancies. In contrast, Mamba3D yields a higher error (0.0197) and a negative $R^2$ (-0.0684), while PointGPT further degrades to a 0.0447 error and $-0.5447$ $R^2$. A negative $R^2$ suggests these methods underperform even a naive "constant mean" predictor in this cross-dataset context. By contrast, DragSolver's robust normalization, multi-scale architecture enable it to retain stable performance despite the substantial shift from DN++ to DML.

**Interior Structure Variation Results.** Industrial pipelines often differ in interior geometry. We model this mismatch by artificially adding random "ghost" interior points to both training and testing data in DrivAerNet++. The training set initially contains only exterior surfaces, but we insert a limited number of synthetic interior points to teach the network that such geometry should be downweighted. At test time, we similarly append more extensive ghost interiors to mimic real scans that include seats or dashboards. Table 6 shows that, without gating, Relative $L_2$ error degrades to 0.078; with gating, the error is reduced to 0.065. Baselines that lack a dedicated filtering mechanism degrade more severely, underscoring the value of surface-guided gating in practical scenarios where training and testing distributions differ in interior geometry.

**Scalability analysis.** As shown in Table 8, **DragSolver** is already the most accurate model with only 5 % of the training data, achieving $R^2 = 0.82$ where the closest baseline (Mamba3D) is still negatively correlated with ground truth. Accuracy improves smoothly as the set grows, yet the gain from 50 % to 100 % data is marginal ($\Delta$ Rel. $L_2 = 0.06 \times 10^{-3}$), indicating good sample–efficiency. The baselines, in contrast, require the full dataset to stabilise and still exhibit 2–6 × higher error at 100 % data. These trends confirm that our architecture scales favourably in *data* and *compute*, delivering state-of-the-art accuracy under limited supervision while adding negligible inference overhead as the dataset expands.

**Noise robustness.** Table 5 shows that **DragSolver** is far less sensitive to real-world–style noise (dropout, pose jitter, sensor error) than the three baselines. Even at the strongest perturbation it keeps a positive $R^2$, whereas PTV3, Mamba3D, and PointGPT all become negatively correlated with CFD ground truth. This confirms the superior resilience of our physics-aware architecture for noisy, in-situ data.

### 4.3. Ablation Study

We now ablate the main modules in three settings: (1) in-distribution, (2) cross-shape, and (3) interior mismatch. The variants tested are: Single-Scale Only: no multi-scale feature fusion.+ Multi-scale (no gating, no normalization). + Multi-scale + Normalization (no gating). DragSolver Full (Multi-scale + Normalization + Gating). Table 7 shows that each module contributes noticeably. Multi-scale is crucial. Comparing "Single-Scale Only" (0.0049 in ID), we see nearly a 47% relative error reduction in the in-distribution scenario. Similar improvements appear in cross-shape (0.0081 vs. 0.0065) and interior mismatch (0.0096 vs. 0.0079), indicating that capturing both global and local geometric cues is essential. Normalization tackles large-scale variation. Adding heterogeneous normalization ("+ MS + Norm.") yields a further jump in accuracy, e.g. from 0.0026 to 0.0007 in the ID setting, showing that unifying vehicle

*Table 5.* Comparison under limited training data (30% split of DrivAerNet++) with increasing augmentation intensity.

| Model | #Train Samples | Time/ epoch (s) | $1\times$ Inf. Time (s) | Drop-out | Rot. | Flip (%) | Trans. | Noise (%) | Rel. $L_2\downarrow$ ($\times 10^{-3}$) | $R^2\uparrow$ |
|---|---|---|---|---|---|---|---|---|---|---|
| DragSolver *(Ours)* | 1638 | 52.09 | 16.82 | 20% | $[-2°\sim2°]$ | 50% | 0.01 | 5% | 2.9420 | 0.8583 |
| | 1638 | 52.02 | 16.81 | 30% | $[-3°\sim3°]$ | 50% | 0.02 | 7% | 5.7857 | 0.7460 |
| | 1638 | 53.46 | 17.51 | 40% | $[-4°\sim4°]$ | 50% | 0.03 | 9% | 16.9077 | 0.2344 |
| PTV3 | 1638 | 42.51 | 14.86 | 20% | $[-2°\sim2°]$ | 50% | 0.01 | 5% | 8.6164 | 0.4669 |
| | 1638 | 43.14 | 14.90 | 30% | $[-3°\sim3°]$ | 50% | 0.02 | 7% | 11.4341 | 0.2501 |
| | 1638 | 43.97 | 15.04 | 40% | $[-4°\sim4°]$ | 50% | 0.03 | 9% | 26.5797 | $-0.5991$ |
| Mamba3D | 1638 | 152.39 | 21.86 | 20% | $[-2°\sim2°]$ | 50% | 0.01 | 5% | 13.7013 | 0.0853 |
| | 1638 | 153.51 | 22.90 | 30% | $[-3°\sim3°]$ | 50% | 0.02 | 7% | 19.8485 | $-0.0894$ |
| | 1638 | 154.54 | 23.04 | 40% | $[-4°\sim4°]$ | 50% | 0.03 | 9% | 37.4115 | $-1.5153$ |
| PointGPT | 1638 | 157.23 | 23.15 | 20% | $[-2°\sim2°]$ | 50% | 0.01 | 5% | 18.3948 | $-0.1941$ |
| | 1638 | 158.32 | 23.45 | 30% | $[-3°\sim3°]$ | 50% | 0.02 | 7% | 36.6032 | $-1.4559$ |
| | 1638 | 159.15 | 24.64 | 40% | $[-4°\sim4°]$ | 50% | 0.03 | 9% | 95.2137 | $-4.8912$ |

*Table 6.* Impact of **interior structure mismatch**: training data has a small portion of synthetic interior points, but test samples include more extensive interior additions. Baseline methods degrade severely when confronted with these unseen interiors, while DragSolver's gating mechanism preserves robust performance.

| Method | Relative $L_2\downarrow$ | $R^2\uparrow$ |
|---|---|---|
| PointTransformer | 0.0096 | 0.4343 |
| PointTransformer V3 | 0.0078 | 0.5455 |
| Mamba3D | 0.0084 | 0.4604 |
| PointGPT | 0.0097 | 0.4241 |
| **DragSolver (No Gating)** | $0.0057\pm 0.0001$ | $0.6538\pm 0.0005$ |
| **DragSolver (Full)** | $\mathbf{0.0024\pm 0.0001}$ | $\mathbf{0.8739\pm 0.0004}$ |

*Table 7.* Ablation of DragSolver components on DrivAerNet++ under (a) in-distribution, (b) cross-shape, and (c) interior mismatch scenarios. Metrics: Relative $L_2$ (lower better).

| Method Variant | (a) In-Dist | (b) Cross-Shape | (c) Interior |
|---|---|---|---|
| Single-Scale Only | 0.0049 | 0.0081 | 0.0096 |
| + Multi-scale | 0.0026 | 0.0065 | 0.0079 |
| + Multi-scale + Norm. | 0.0007 | 0.0034 | 0.0057 |
| **DragSolver (Full)** | $\mathbf{0.0007\pm 0.0001}$ | $\mathbf{0.0034\pm 0.0002}$ | $\mathbf{0.0024\pm 0.0002}$ |

*Table 8.* Scalability analysis on DrivAerNet++ under different training-set sizes. Lower relative $L_2$ is better; higher $R^2$ is better.

| Model | #Train Samples | Train Ratio | Time/ epoch (s) | $1\times$ Inf. Time (s) | Rel. $L_2\downarrow$ ($\times 10^{-3}$) | $R^2\uparrow$ |
|---|---|---|---|---|---|---|
| DragSolver | 268 | 5% | 5.40 | 11.98 | 3.5119 | 0.8243 |
| | 536 | 10% | 12.62 | 12.04 | 1.3929 | 0.9255 |
| | 1638 | 30% | 37.54 | 12.14 | 0.7026 | 0.9620 |
| | 2680 | 50% | 60.73 | 12.68 | 0.5927 | 0.9673 |
| | 5361 | 100% | 124.50 | 13.25 | 0.5351 | 0.9710 |
| PTV3 | 268 | 5% | 4.48 | 11.54 | 68.1518 | $-3.0564$ |
| | 536 | 10% | 9.74 | 11.40 | 66.7872 | $-2.8737$ |
| | 1638 | 30% | 29.07 | 11.41 | 6.4185 | 0.6150 |
| | 2680 | 50% | 48.81 | 11.86 | 3.2661 | 0.8088 |
| | 5361 | 100% | 98.77 | 12.25 | 1.7319 | 0.9024 |
| Mamba3D | 268 | 5% | 17.02 | 22.03 | 42.1424 | $-1.6104$ |
| | 536 | 10% | 35.12 | 22.15 | 32.8624 | $-1.1419$ |
| | 1638 | 30% | 103.54 | 22.30 | 9.2257 | 0.3926 |
| | 2680 | 50% | 178.41 | 22.30 | 2.8901 | 0.8362 |
| | 5361 | 100% | 343.65 | 23.01 | 1.4811 | 0.9191 |
| PointGPT | 268 | 5% | 18.21 | 23.54 | 70.8708 | $-3.8234$ |
| | 536 | 10% | 37.02 | 23.64 | 62.9200 | $-3.2949$ |
| | 1638 | 30% | 106.32 | 23.41 | 13.2661 | 0.1315 |
| | 2680 | 50% | 185.15 | 23.56 | 6.2985 | 0.6180 |
| | 5361 | 100% | 369.01 | 23.67 | 3.1240 | 0.8207 |

*Table 9.* Impact of Multi-Scale Depth $\boldsymbol{L}$ on DrivAerNet++.

| Depth | L=1 | L=3 | L=5 | L=7 |
|---|---|---|---|---|
| Relative $L_2\downarrow$ | 0.0020 | 0.0010 | **0.0008** | 0.0009 |

sizes/densities helps the model converge more effectively. Gating is pivotal for interior mismatch. In the final column, adding gating drives error from 0.0057 to 0.0024, a 58% reduction, clearly demonstrating how surface-guided filtering helps when test data includes unexpected internal structures.

## 4.4. Model Analysis

We further analyze core hyperparameters and design decisions in **DragSolver**, focusing on DrivAerNet++ under the in-distribution (ID) scenario unless stated otherwise. Our goal is to demonstrate how multi-scale depth, pooling strategy, voxel granularity, sampling approaches, and runtime efficiency collectively shape the final performance.

**Multi-Scale Depth** ($L$). As discussed in Sec. 3.2, we vary the encoder depth $L \in \{1, 3, 5, 7\}$ to assess how capturing multiple scales influences accuracy. Table 9 indicates that $L = 5$ achieves the lowest relative $L_2$ error (0.0008). When $L = 1$, the network struggles to encode global shape context (error 0.0020), and $L = 7$ brings diminishing returns with higher overhead. Hence, $L = 5$ balances accuracy and efficiency.

**Global Pooling: Max vs. Average.** After each encoder stage (Sec. 3.2), we apply a global pooling operator (GPool). As shown in Table 10, max pooling consistently

Table 10. Impact of Global Pooling.

| Global Pooling Strategies | Mean | Max |
|---|---|---|
| Relative $L_2 \downarrow$ | 0.0009 | **0.0007** |

Table 11. Impact of voxel size $\delta$ (lower $\delta$ = finer granularity).

| Voxel Size | 0.01 | 0.05 | 0.1 |
|---|---|---|---|
| Relative $L_2 \downarrow$ | **0.0009** | 0.0016 | 0.0029 |

outperforms average pooling, likely due to its ability to better retain salient geometric features (e.g., spoilers, ridges), which are crucial for aerodynamic prediction.

**Voxel Size.** When using a sparse voxel backend, we evaluate voxel sizes $\delta \in \{0.01, 0.05, 0.10\}$ in normalized coordinates. As Table 11 suggests, smaller $\delta = 0.01$ captures more fine-grained geometry, improving accuracy but slightly increasing memory/time cost. Larger voxels ($\delta = 0.10$) lose local detail, hurting $C_d$ prediction. We thus adopt $\delta = 0.01$.

**Hybrid Sampling.** To handle heterogeneous vehicle meshes, we compare three sampling strategies under an OOD setting on DrivAerNet++. Specifically, the test distribution includes shapes and resolutions not seen at training. Mesh-based sampling alone can overfit to the training mesh topology, while uniform sampling may underrepresent high-curvature regions. Table 12 shows that a hybrid approach achieves the best accuracy, effectively balancing local geometric detail against distribution shifts.

**Point Downsampling.** Table 13 quantifies the impact of progressively reducing the number of surface points fed to the network. A gentle reduction from 60 k to 40 k points increases the relative $L_2$ by only $\sim 0.07! \times !10^{-3}$ and keeps $R^2$ above 0.95, indicating that the model is robust to moderate sparsity. Below 30 k points, however, errors rise rapidly—most notably a six-fold jump in $L_2$ and a 19-point drop in $R^2$ at 10 k points—showing that excessive decimation removes critical boundary-layer information.

Table 12. Comparing sampling strategies.

| Strategies | mesh-based | uniform | hybrid |
|---|---|---|---|
| Relative $L_2 \downarrow$ | 0.0019 | 0.0016 | **0.0012** |

**Training and Inference Efficiency.** Finally, we compare DragSolver's efficiency against state-of-the-art 3D point-based methods under the same hardware environment. As highlighted in PTV3 (Wu et al., 2024b), KNN-based approaches (e.g., PointGPT (Chen et al., 2024), Mamba3D (Han et al., 2024)) incur high computational costs due to repeated neighbor graph construction and attention computation. Inspired by this, we adopt a SpConv-based sparse convolutional backbone (Contributors, 2022)

Table 13. Performance evaluation of DragSolver under varying numbers of sampled points.

| #Train Samples | #Sampled Points | Sampled Pts. Ratio | Rel. $L_2 \downarrow$ ($\times 10^{-3}$) | $R^2 \uparrow$ |
|---|---|---|---|---|
| 1638 | 60 000 | 2.72 – 14.38% | 0.7026 | 0.9620 |
| 1638 | 50 000 | 2.27 – 11.99% | 0.7442 | 0.9606 |
| 1638 | 40 000 | 1.18 – 9.59% | 0.7766 | 0.9572 |
| 1638 | 30 000 | 1.36 – 7.19% | 0.8844 | 0.9534 |
| 1638 | 20 000 | 0.59 – 4.79% | 1.0284 | 0.9460 |
| 1638 | 10 000 | 0.30 – 2.40% | 4.1502 | 0.7718 |

combined with FlashAttention (Dao, 2024), significantly reducing computational complexity by avoiding costly KNN operations. Table 14 shows that DragSolver achieves 3× faster training time compared to Point-based methods. Our method processes normalized point in under 0.9 s with $L = 5$, $\delta = 0.01$. Monte Carlo Dropout ($T = 10$) raises inference to ≈4.8 s/shape, yet remains far below CFD times (hours to days), thereby retaining viability for rapid design.

Table 14. Training/inference time (per epoch, per sample).

| Method | PointGPT | Mamba3D | PTV3 | Ours |
|---|---|---|---|---|
| Train (s/epoch) | 1680 | 1600 | **490** | 530 |
| Infer (s/sample) | 0.97 | 0.95 | **0.79** | 0.93 |

## 5. Conclusions and Feature Work

We presented DragSolver, a Transformer-based framework for real-time drag coefficient estimation from 3D automotive geometries. By integrating multi-scale features, heterogeneous normalization, surface-guided gating, and MC Dropout uncertainty, DragSolver robustly handles diverse mesh densities, interior structures, and shape variations. Experiments across three real-world datasets show state-of-the-art results and near real-time inference, confirming each module's contribution and the method's potential for practical automotive design. Future work will explore partial CFD constraints, expanded uncertainty calibration, and broader domains (e.g. aerospace).

## Acknowledgements

This work was supported by the National Key Research and Development Program of China 2024YFF1500600, and the Innovation Capability Support Program of Shaanxi(Program No.2023-CX-TD-30). We also thank *Shenzhen TenFong Technology Co. Ltd.* for valuable engineering support throughout the project.

## Impact Statement

Our approach accelerates aerodynamic evaluation, potentially reducing prototyping costs and improving energy efficiency. While we do not foresee unusual ethical issues, biased or incomplete training data could yield unexpected failure on certain vehicle shapes. Proper dataset diversity and real-world validation remain critical. This paper aims to advance Machine Learning for automotive aerodynamics. We identify no major ethical or societal risks beyond standard ML considerations.

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

## A. Visualization of Predicted Drag Coefficients

In this section, we present additional qualitative insights into DragSolver's predictions on various car geometries. Specifically, we visualize each vehicle's exterior mesh (or point cloud) alongside its predicted drag coefficient ($C_d$) and compare it to the ground-truth reference (e.g., wind-tunnel or CFD-derived). Figure 4 illustrates the predicted and ground truth drag coefficients ($C_d$) across multiple vehicle geometries. The model demonstrates high accuracy, with predictions closely aligning with ground truth values. The results underscore the effectiveness of our framework in learning aerodynamic features and estimating drag coefficients in real-world automotive applications.

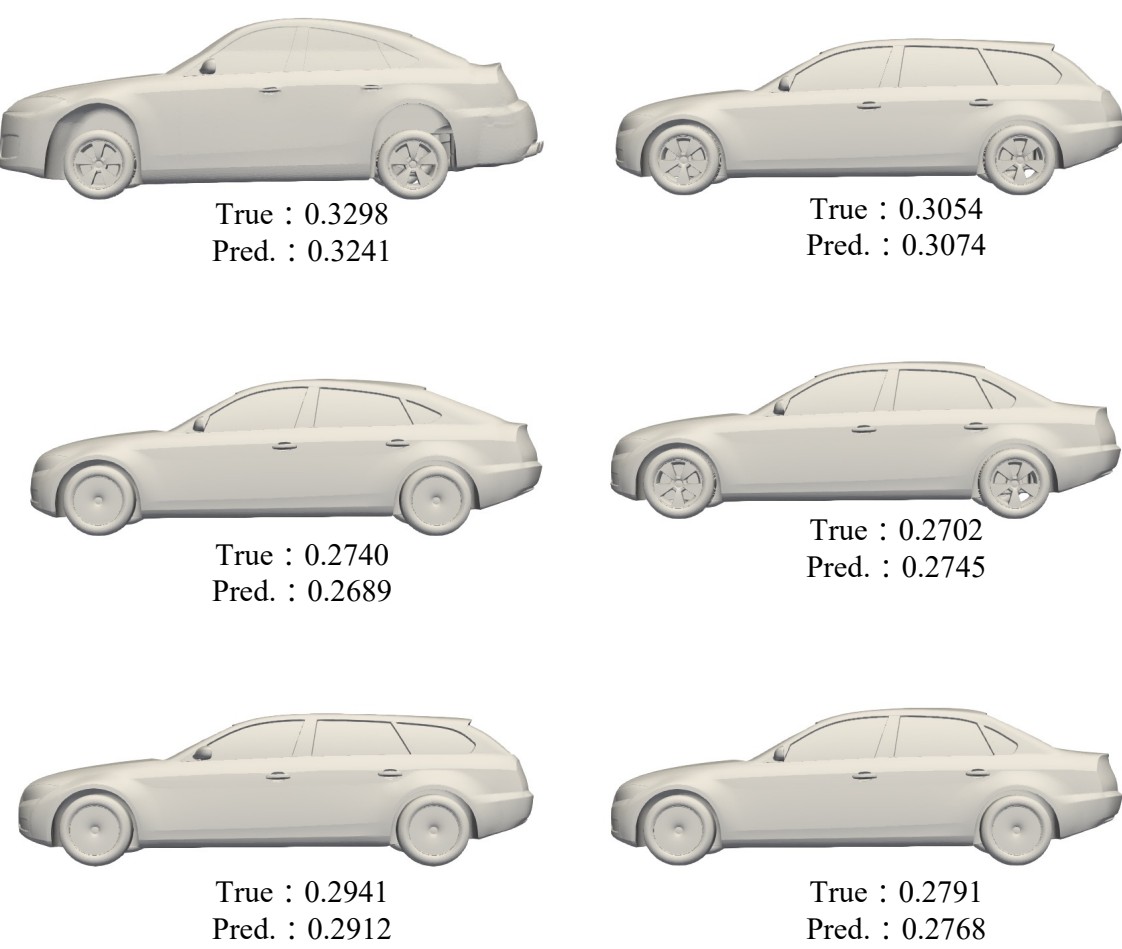

*Figure 4.* Comparison between true and predicted aerodynamic drag coefficients ($C_d$) for different vehicle geometries. Each row presents a pair of vehicles with their corresponding ground truth and predicted $C_d$ values. The top row shows vehicle designs with higher drag values (**True:** $C_d = 0.3298, 0.3054$; **Pred:** $C_d = 0.3241, 0.3074$), indicating greater aerodynamic resistance. The middle row illustrates models with moderate drag coefficients (**True:** $C_d = 0.2740, 0.2702$; **Pred:** $C_d = 0.2689, 0.2745$). The bottom row presents the most aerodynamically optimized designs (**True:** $C_d = 0.2941, 0.2791$; **Pred:** $C_d = 0.2912, 0.2768$). The results demonstrate the accuracy of our model in predicting $C_d$, with minimal deviation from the true values. Notably, variations in aerodynamic design lead to distinct drag values, impacting energy efficiency in real-world applications.

## B. Detailed Dataset Descriptions

**Overview of Datasets.** Figure 5 offers a comparative look at the three automotive datasets used in our experiments: **DrivAerNet++**, **DrivAerNet**, and **DrivAerML**. Each varies in mesh density, inclusion of internal geometry, and parametric scope. Such differences demand robust mesh normalization and multi-scale feature extraction (as detailed in our main text) to achieve reliable $C_d$ regression. Below, we provide a concise, third-person summary of each dataset, referencing the

original authors and publications.

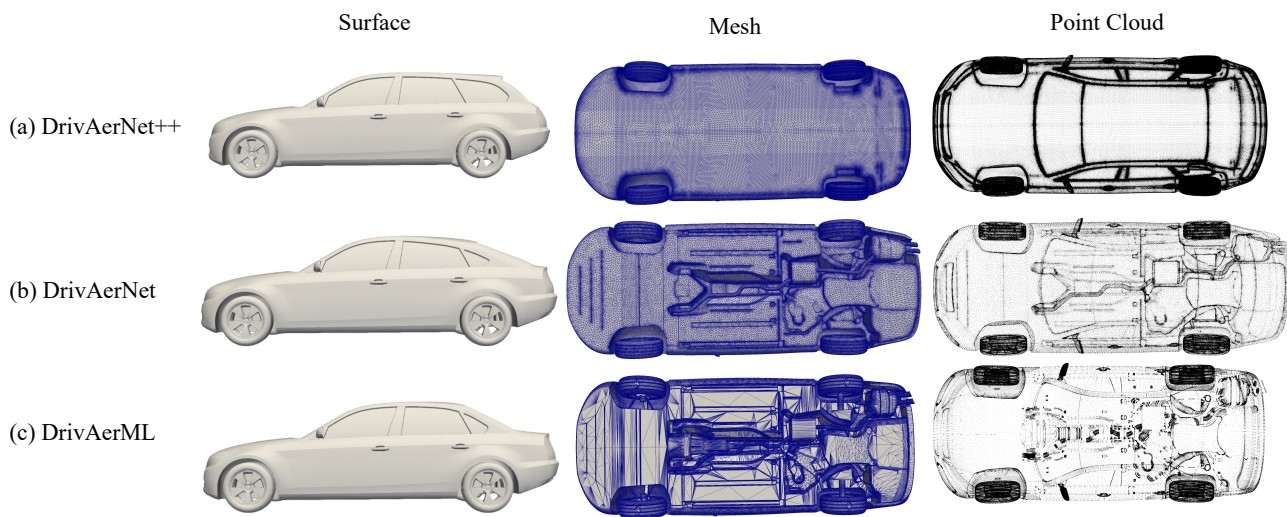

Surface        Mesh        Point Cloud

(a) DrivAerNet++

(b) DrivAerNet

(c) DrivAerML

*Figure 5.* **Visual Comparison of DrivAerNet++, DrivAerNet, and DrivAerML.** Each row corresponds to a dataset: **(a)** DrivAer-Net++ (high-fidelity external surfaces, parametric), **(b)** DrivAerNet (conventional automotive meshes capturing underbody details), **(c)** DrivAerML (even denser CAD scans including internal structures). For each dataset, we show the surface geometry (left), mesh representation (middle), and a point-cloud view (right). The progression from simpler external shapes to highly detailed, partially interior meshes underscores the increasing complexity and data richness across the datasets. These contrasting resolutions, structural details, and geometric variations pose challenges for generalization in $C_d$ prediction.

## B.1. DrivAerNet++ Dataset

**Origin and Motivation.** Elrefaie et al. (2024b) introduced *DrivAerNet++* to supply a large-scale collection of watertight 3D car meshes, validated against the DrivAer model (Heft et al., 2012b;a). Unlike generic repositories (e.g. ShapeNet (Chang et al., 2015)), DrivAerNet++ aims to preserve engineering-quality resolution for aerodynamic simulations.

**Parametric Models.** Using ANSA® morphing, up to 26 design parameters alter both global car dimensions (length, roof height) and local features (diffusor, underbody, wheels). This yields baseline configurations—*fastback, estateback, notchback*—expanded to thousands of variants, each with a final watertight mesh. Body shapes differ significantly in rear geometry, occupant space, , matching diverse industrial needs.

**CFD Simulation and Validation.** Elrefaie et al. (2024b) use `snappyHexMesh` (OpenFOAM Foundation, 2023) to refine around wheels, underbody, and wake regions, often reaching 12–24 million cells. They adopt a $k$–$\omega$ SST RANS approach at $30 \, \mathrm{m/s}$ (Re$\approx 10^7$), achieving drag predictions within 2–5% of TUM's wind-tunnel data (Heft et al., 2012a). While the fidelity is lower than full LES, it balances mesh size vs. feasibility for large parametric sweeps.

**Relevance for ML.** Because DrivAerNet++ covers *multiple* rear-end shapes (fastback, estateback, notchback) at moderate to high mesh resolution, it provides a diverse design space for training data-driven aerodynamic surrogates. The authors note possible future expansions with multi-fidelity or hybrid RANS-LES approaches.

## B.2. DrivAerNet Dataset

**Background and Baseline Geometry.** Elrefaie et al. (2024a) present *DrivAerNet* as an earlier, flexible variant of the DrivAer model (Heft et al., 2012b), bridging simple bodies (Ahmed (Ahmed et al., 1984) or SAE (Cogotti, 1998)) and proprietary industrial CAD. Unlike the single baseline STL, they introduced about 50 morphable parameters (roof angle, wheel/mirror geometry) to broaden shape variation.

**CFD Setup and Mesh.** In DrivAerNet, the authors also rely on OpenFOAM® with a $k$–$\omega$ SST scheme, refining the underbody, wheels, and wake regions via `snappyHexMesh`. A typical velocity is $30 \, \mathrm{m/s}$, matching the TUM reference, and the Re$\approx 9.39 \times 10^6$. They particularly highlight a fastback with detailed underbody + wheels + mirrors (FDwWwM),

since each adds significant drag counts. Summaries show that these morphological changes are well captured within a few percent of experimental values.

**Parameter Range and Mesh Resolution.** Mesh resolutions vary from around 500k faces up to millions of cells, enabling more detailed geometry than prior open datasets (Li et al., 2023a; Song et al., 2023). The authors provide code and morphological scripts at `https://github.com/Mohamedelrefaie/DrivAerNet/ParametricModel`, offering an external resource for data-driven aerodynamic tasks.

### B.3. DrivAerML Dataset

**Notchback Configuration.** DrivAerML around the notchback variant of the DrivAer body, originally used in the 2nd–4th AutoCFD Workshops. Wheels are static (no rotation), and cooling inlets are sealed to reduce complexity and facilitate consistent flow-field comparisons among different CFD codes.

**CFD Domain and Mesh.** They employ a large "open-road" domain with negligible blockage ratio ($\sim 0.25\%$). The free-stream velocity is $38.889 \, \text{m/s}$, giving Re$\approx 7.19 \times 10^6$ based on wheelbase $L = 2.786 \, \text{m}$. The reference frontal area $A = 2.17 \, \text{m}^2$ is used for force and moment coefficients. Mesh generation uses ANSA's HeXtreme, producing a hexa/polyhedral grid of $\sim 160$ million cells (high-$y^+$ approach, 7 prism layers). This strategy, validated in the 2nd AutoCFD workshop, balances resolution and runtime for industrial-level setups, capturing underbody flow, mirrors, and wake zones.

**Focus on Industrial Fidelity.** DrivAerML aims to provide a near-industrial RANS mesh quality—particularly relevant for advanced turbulence modeling or data-driven aerodynamic surrogates. The geometry's sealed inlets and static wheels reduce complexity yet maintain key automotive features (e.g. underbody structures, detailed external surfaces). Full details, domain boundaries.

**Conclusion.** These three datasets—DrivAerNet++, DrivAerNet, and DrivAerML—collectively span a wide spectrum of vehicle shapes, mesh densities, and structural details (exterior-only vs. partial interior). Each dataset has been validated against DrivAer wind-tunnel data from TUM or other references, making them valuable external resources for learning-based $C_d$ prediction and surrogate modeling of aerodynamic flows.

## C. Implementation Details

This appendix provides additional information about DragSolver's implementation and the experimental protocols referenced in §4. It covers model hyperparameters, training configurations, hardware setups, and specific differences from baseline methods.

### C.1. Model Configuration

**Overall Architecture.** As described in §3.2, DragSolver uses a *five-stage* encoder, each halving the spatial resolution (stride = 2) via sparse convolutions or point-based downsampling. For each stage $k$, the channel dimension grows from an initial 32 to up to 256 by the final stage (Table 15). We adopt multi-head attention with $H = 8$ heads. A patch-based or voxel-based partition is used to handle large unstructured point sets. Following §3.3, we fix the wheelbase to $2.8 \, \text{m}$, then apply a hybrid sampling (uniform + mesh-based) that yields $\sim$600k effective points per shape.

**Surface-Guided Gating.** We apply the depth-based filtering from §3.4 at an initial voxel resolution of $\Delta = 0.01$ m. Voxels beyond a threshold depth (e.g. 40% layers inside) are tagged internal and removed. In high-resolution input layers, each surviving voxel has an initial gating weight $c_i = 1$ and is refined by a gating MLP ($\text{MLP}_{\text{input}}$). We skip gating at intermediate layers, then re-apply gating at the final layer via $\text{MLP}_{\text{final}}$.

**Uncertainty Estimation.** We use MC Dropout (Gal & Ghahramani, 2016) with a dropout rate of 0.1 in the final regression MLP. At inference, each shape is passed $K = 10$ times with different dropout masks, yielding an ensemble $\{\hat{C}_d^{(1)} \ldots \hat{C}_d^{(10)}\}$. The mean $\overline{C}_d$ is reported as the prediction, while variance approximates epistemic uncertainty.

*Table 15.* Model layer configuration for DragSolver, with 5 stages plus a final regression MLP. Notation: $d_{\text{in}}$ is input channels, $d_{\text{out}}$ is output channels. After concatenating multi-scale features into a 768-dimensional vector, we feed it into the shown MLP structure to predict $C_d$.

| Stage | Input Size | # Blocks | #Heads | Output Channels |
|---|---|---|---|---|
| Embedding | 3 | 1 | – | 32 |
| Stage 1 | 32 | 2 | 2 | 64 |
| Stage 2 | 64 | 2 | 4 | 128 |
| Stage 3 | 128 | 2 | 8 | 128 |
| Stage 4 | 128 | 4 | 16 | 256 |
| Stage 5 | 256 | 2 | 16 | 256 |
| **Regression** | $\underline{\text{Concat } \mathbf{g}^{(0)}\|\mathbf{g}^{(1)}\|\cdots\|\mathbf{g}^{(5)} \to \mathbb{R}^{768}}$ $\xrightarrow{\text{Linear}(768\to256)} \xrightarrow{\text{BN}} \xrightarrow{\text{ReLU}} \xrightarrow{\text{Dropout}} \xrightarrow{\text{Linear}(256\to128)} \xrightarrow{\text{BN}} \xrightarrow{\text{ReLU}} \to \text{Linear}(128\to1)$ | | | |

## C.2. Training Configuration

**Training Hyperparameters.** Unless otherwise stated, we train for 50 epochs using AdamW (Loshchilov, 2019) with a base learning rate of $1 \times 10^{-3}$ and a cosine decay schedule. Our batch size is 24 (due to memory constraints). We set weight decay to $5 \times 10^{-5}$. We do not tune these hyperparameters across tasks, using a unified setting for consistency.

**Losses.** As described in §3.1, we define a standard MSE loss on the predicted $C_d$. Specifically,

$$\mathcal{L}_{\text{reg}} = \frac{1}{M}\sum_{j=1}^{M}\big(\hat{C}_d^{(j)} - C_d^{(j)}\big)^2.$$

**Data Preprocessing.** For DrivAerNet++ and DrivAerNet, we reserve 20% for testing. We randomize the order of shapes each epoch. For cross-shape or cross-dataset evaluations, we isolate the geometry categories or entire dataset for training vs. testing as described in §4.1.

## C.3. Comparison with Baselines

**Point-based Methods.** We implement or take code from offical library for standard 3D methods: PointNet, PointNet++, Mamba3D. Each uses the recommended AdamW or Adam optimizer with identical learning rate as DragSolver. Graph-based expansions (e.g. DGCNN, MeshGraphNet) require adjacency or face definitions, which are straightforward in DrivAerNet++ but require additional conversion for DrivAerML's partial interior data. We attempt to keep their hyperparameters near the official defaults, only adjusting channel size and training epochs to match our setup.

**Point-based Methods (Baselines).** We adopt or reference the official code repositories from the original authors aiming to remain as faithful as possible to their recommended training procedures. For consistency with DragSolver's approximate 600 k-point representation (see §3.3), we also sample 600 k points for each baseline method. This ensures all models operate on similarly detailed point sets and fairly compare performance on high-density automotive meshes. Because many of these baselines rely on $k$-nearest neighbors (KNN) for local feature aggregation, we fix $k = 32$ to capture sufficient neighborhood context in complex car surfaces. However, this large $k$ significantly increases memory usage, nearly exhausting GPU resources in our setup. Apart from adjusting the batch size to accommodate memory constraints, we keep other hyperparameters as suggested by the original authors (e.g., learning rates, layer widths, activations).

**Neural Operators / PDE Solvers.** We adapt PDE-based surrogates (e.g. GNOT, Transolver (Wu et al., 2024a)) by letting them predict a single scalar $C_d$. That is, instead of local field predictions, we pool features globally and feed an MLP to regress $C_d$.

## C.4. Hardware and Runtime Statistics

We conduct all experiments on up to 2 *NVIDIA A800* GPUs. On DrivAerNet++ at full resolution ($\sim 600k$ points), DragSolver typically processes a batch of size 24. Inference on a single shape is under 100 ms for a single forward pass, plus $K \times 100$ ms for MC Dropout. This is dwarfed by the hours or days needed for high-fidelity CFD or wind tunnel tests.

## C.5. License and Data Availability.

DrivAerNet++ and DrivAerNet are from Elrefaie et al. (2024a), while DrivAerML is from Ashton et al. (2024). Please see Appendix B for direct links and usage instructions.

