# OpenReview forum: "DragSolver: A Multi-Scale Transformer for Real-World  Automotive Drag Coefficient Estimation"
_ICML.cc/2025/Conference — ICML 2025 poster_

### Official Review · Reviewer_pidM · 2025-03-11

**Overall Recommendation:** 2

**Summary:**

The authors present DragSolver, a Multi-scale Transformer for processing car point clouds to estimate drag coefficient for automotive designs in real-world applications. To adapt traditional transformer architectures to this new task, the authors propose multiple designs to achieve more trustful results, like 1) multi-scale feature extraction, 2) heterogeneous scale normalization to process vehicles in diverse sizes, 3) surface guided gating to alleviate distractions caused by irrelevant interior designs, and 4) Monte Carlo output drop layer to  estimate the range of drag coefficient for real-world applications. The results show the superiority of the architecture design.

**Claims And Evidence:**

The presentation is clear.

**Essential References Not Discussed:**

All related works are included.

**Experimental Designs Or Analyses:**

The experiments are well-designed. The ablations are sufficient.

**Methods And Evaluation Criteria:**

1. In line 60, the author emphasize that both local details and global shape influence drag. Why do the authors choose point cloud over mesh as the model input? The point cloud is not always an efficient 3D representation as it requires a vast number of points to capture high-frequency patterns like sharp edges. Also, reconstructing meshes from point clouds leads to artifacts, indicating that point cloud might not be an ideal way that preserves sufficient local feature for estimating drag.

2. In line 210, the authors normalize the sizes of different vehicles by normalizing the wheelbase length to a fixed number. However, for real-world applications, the wheelbase length is not always standardized, *e.g.* applying a fixed wheelbase length for `Ford F-450` and `Peel P50` surely leads to an unexpected result. Therefore, why not directly use their actual size in the real world?

3. In line 264 (Sec. 3.5), the authors propose MC Dropout for the output layer (Module 4 in Fig. 2) for diverse estimations to provide an error interval estimation. Dropout is mostly a technique adopted during training to prevent overfitting. Can this be accomplished with a point cloud sampling method to produce diverse inputs?

**Other Comments Or Suggestions:**

None

**Other Strengths And Weaknesses:**

None

**Questions For Authors:**

None

**Relation To Broader Scientific Literature:**

This paper is closely related to point cloud processing methods like PointNet, PointNet++, ... as this paper focuses on designing 3D point cloud operators to estimate drag coefficients for vehicles. The paper is also closely related to DrivaerNet, DrivaerNet++, DrivaerML as they all focus on estimating drag coefficients.

**Theoretical Claims:**

The theoretical claims are well-addressed.

---

> ### Author Rebuttal · Authors · 2025-03-31
>
> We sincerely appreciate the reviewer’s insightful and constructive comments. We have conducted additional experiments and analyses based on your suggestions, and these results will be explicitly included in the revised manuscript.
>
> **Important Note:**  To clearly demonstrate the thoroughness of these additional experiments, detailed experimental configurations are provided in [**Table 1 (Click to View)**](https://anonymous.4open.science/r/ICMLRebuttal-F329/Table%201.jpg). Additional results mentioned in our responses are provided anonymously here:  [**Anonymous Supplementary Tables**](https://anonymous.4open.science/r/ICMLRebuttal-F329)
>
> ---
>
> **Q1: Point Clouds vs. Meshes**
>
> **A1:** We appreciate the reviewer’s insightful comment regarding the choice of point clouds over meshes. Indeed, mesh-based methods can effectively encode surface connectivity and local geometric details, but their performance heavily depends on the quality and consistency of mesh discretization. Achieving high-quality mesh generation often requires substantial domain expertise and careful tuning specific to each geometric structure, significantly increasing complexity and cost. To ensure general applicability, flexibility, and ease of use—especially for large-scale automotive datasets—we adopted point clouds. Point clouds naturally bypass issues related to mesh discretization and topology, simplify data processing pipelines, and enable straightforward implementation of various data augmentation techniques (e.g., random sampling, rotation, noise addition), enhancing robustness and generalization. Moreover, to address your concern about capturing local geometric details, we explicitly evaluated DragSolver's accuracy across different point cloud sampling densities ([**Table 4**](https://anonymous.4open.science/r/ICMLRebuttal-F329/Table%204.jpg)). Our experiments confirm that DragSolver maintains strong predictive accuracy even with significantly reduced point counts (as few as 10,000 points), effectively capturing both local and global aerodynamic features necessary for accurate drag estimation. We will explicitly clarify this rationale in the revised manuscript.
>
> **Q2: Wheelbase Normalization**
>
> **A2:** We acknowledge the reviewer’s concern regarding normalization of wheelbase length. Although directly using actual vehicle dimensions is intuitive, significant scale differences (e.g., Ford F-450 vs. Peel P50) can cause substantial training instability and poor generalization, especially with limited or noisy training data. Our comparative experiments ([**Table 5**](https://anonymous.4open.science/r/ICMLRebuttal-F329/Table%205.jpg)) clearly demonstrate that training without wheelbase normalization results in notably higher errors and instability (e.g., Relative *L²*: 0.0057 without normalization vs. 0.0014 with normalization, under limited 30% training data conditions). Importantly, our normalization strategy uniformly scales vehicle geometries to a consistent reference length without altering their inherent shapes, thus significantly reducing scale-related instability and improving predictive accuracy. In practical scenarios, predictions can be effortlessly rescaled back to the actual dimensions of the vehicles. We will explicitly clarify this point in the revised manuscript.
>
> **Q3: MC Dropout vs. Point Cloud Sampling**
>
> **A3:** We appreciate the reviewer’s insightful comment. Indeed, Dropout is commonly employed during training to prevent overfitting. However, in our method, MC Dropout is specifically utilized during inference as an approximate Bayesian approach to estimate epistemic (model) uncertainty [1], which differs fundamentally from the aleatoric (data) uncertainty captured by random point cloud sampling. To clearly illustrate this distinction, we conducted additional comparative experiments ([**Table 6**](https://anonymous.4open.science/r/ICMLRebuttal-F329/Table%206.jpg)). The results show that random sampling alone primarily addresses aleatoric uncertainty and thus cannot adequately capture epistemic uncertainty, which MC Dropout specifically targets. Moreover, combining both methods provides a balanced estimation of both uncertainties, leading to a more robust prediction. We will explicitly clarify this distinction in the revised manuscript.
>
> [1] Gal, Yarin, and Zoubin Ghahramani. "Dropout as a Bayesian Approximation: Representing Model Uncertainty in Deep Learning." *ICML*. PMLR, 2016.

---

> > ### Comment · Reviewer_pidM · 2025-04-07
> >
> > I thank the authors for their detailed feedback, which has addressed some of my concerns.
> >
> > I understand that point cloud is a widely-adopted method to represent 3D shapes as it can be processed with lower cost and complexity. Regarding the point cloud representation, I believe certain techniques like mesh surface importance sampling might compensate for the information loss caused by transforming meshes into point clouds. However, as for the data augmentation part, certain augmentation techniques like `rotation`, `noise addition` do not seem appropriate, as they will surely affect the drag coefficients.
> >
> > As is shown in [Figure 6](https://anonymous.4open.science/r/ICMLRebuttal-F329/Table%206.jpg), the combination of both `model and data uncertainty` leads to the largest estimation variance for DragSolver, but why is its estimation variance smaller than `model uncertainty only` for PTv3 (*e.g.* $\pm$0.1431 v.s. $\pm$0.1647 for Relative $L^{2}$)?

---

> > > ### Author Response · Authors · 2025-04-08
> > >
> > > We sincerely appreciate the reviewer’s insightful and constructive comments. We respond in detail below, referencing additional experimental evidence ([**Table 7**](https://anonymous.4open.science/r/ICMLRebuttal-F329/Table%207.jpg)) to clarify the concerns raised.
> > >
> > > **Q1: Data Augmentation and Its Impact on Drag Coefficients**
> > >
> > > **A1:** We fully agree with the reviewer’s point that certain augmentation techniques, such as rotation and noise addition, could indeed be inappropriate because they may significantly affect the drag coefficients. In fact, if all available data were perfect, we would prefer to avoid using data augmentation altogether. However, real-world data are rarely perfect and often contain minor perturbations, such as slight misalignments during scanning, partial occlusions, or small manufacturing variations. Thus, our data augmentation strategy—small-angle rotations (±2–4°), mild translations, and moderate noise additions—is deliberately designed to simulate these realistic imperfections, allowing us to evaluate the model’s robustness under practical conditions. We strictly avoid large-scale transformations (e.g., 90° rotations) that would invalidate the physical meaning of the drag coefficient. In the revised manuscript, we will explicitly clarify this motivation. As shown in [**Table 3**](https://anonymous.4open.science/r/ICMLRebuttal-F329/Table%203.jpg), our model exhibits superior robustness compared to other models, consistently maintaining stable performance under these controlled augmentations, further validating its resilience to real-world noise and imperfections.
> > >
> > > **Q2: Uncertainty Comparison: DragSolver vs. PTv3**
> > >
> > > **A2:** We greatly appreciate the reviewer’s careful observation regarding uncertainty behavior presented in [**Table 6**](https://anonymous.4open.science/r/ICMLRebuttal-F329/Table%206.jpg). Indeed, DragSolver exhibits an additive effect when combining aleatoric and epistemic uncertainties, resulting in the highest overall variance. Conversely, for PTv3, combined uncertainty is lower than epistemic uncertainty alone, indicating a complementary effect.
> > >
> > > To better understand and clarify this phenomenon, we performed additional experiments detailed in [**Table 7**](https://anonymous.4open.science/r/ICMLRebuttal-F329/Table%207.jpg). These supplementary results clearly indicate that the interactions between aleatoric (data) and epistemic (model) uncertainties are significantly influenced by model architecture and random seed configurations, rather than being specific to a particular model architecture:
> > >
> > > - For DragSolver, the interaction between aleatoric and epistemic uncertainties is significantly influenced by the choice of random seed. Specifically, for Seed=1 and Seed=5, we observed an additive effect, meaning these uncertainty sources independently increase the overall variance. In contrast, with Seed=3, DragSolver exhibits a complementary (smoothing) effect, whereby aleatoric uncertainty mitigates fluctuations caused by epistemic uncertainty, effectively reducing the total variance.
> > >
> > > - Conversely, PTv3 generally demonstrates complementary behavior with Seed=1 and Seed=5, exhibiting a stabilizing effect. However, the influence of the random seed is evident with Seed=3, where PTv3 shows an additive effect, as both uncertainty types independently contribute to increasing the total variance.
> > >
> > > Importantly, this variability is **not a weakness**, but rather reflects meaningful differences caused by the model architectures and random seed selections.
> > >
> > > >**Specifically, the interaction between aleatoric (data-related) and epistemic (model-related) uncertainties is not fixed. Instead, it changes according to different random seeds, highlighting how sensitive the uncertainty estimation is to stochastic factors.**
> > >
> > > >**Rather than indicating a lack of stability, this variability helps us better understand how different model architectures and the randomness inherent in data sampling together influence uncertainty estimates.**
> > >
> > > By conducting experiments across multiple random seeds, we obtain a deeper and more robust understanding of each model’s stability and the relative contributions of different uncertainty types. We will explicitly clarify this explanation in the revised manuscript to avoid confusion and better emphasize the significance of our uncertainty analysis.
> > >
> > > Again, thank you for your insightful comments. These points significantly strengthen the clarity and rigor of our manuscript.

---

### Official Review · Reviewer_mQmd · 2025-03-13

**Overall Recommendation:** 2

**Summary:**

The paper presents a Transformer-based framework designed for predicting the aerodynamic drag coefficient of automotive designs directly from 3D vehicle models. This work is motivated by the high computational costs and inefficiencies of traditional Computational Fluid Dynamics (CFD) simulations and wind tunnel experiments, which, despite their accuracy, are often too slow for rapid design iterations. The authors propose DragSolver as a deep learning-based surrogate model that integrates multi-scale feature extraction, heterogeneous scale normalization, surface-guided gating, and epistemic uncertainty estimation to improve the reliability and generalizability of aerodynamic predictions.

**Claims And Evidence:**

One of the strengths of the paper is the multi-scale feature extraction mechanism, which effectively captures both global shape characteristics and fine-grained local geometric details that influence aerodynamics. Another notable contribution is the surface-guided gating mechanism, which suppresses irrelevant internal structures such as seats and dashboards that are often present in 3D scans but do not impact external aerodynamic behavior.

**Essential References Not Discussed:**

None.

**Experimental Designs Or Analyses:**

The paper compares perfomrance on 3-different datasets.

**Methods And Evaluation Criteria:**

The paper just used fully supervised training. It does not provide a detailed computational efficiency analysis in comparison to alternative CFD-based surrogate models

**Other Comments Or Suggestions:**

None

**Other Strengths And Weaknesses:**

Although DragSolver is significantly faster than traditional CFD simulations, the inference time of 0.9 to 5 seconds per shape could still be a limitation in real-time design applications where instant feedback is necessary. A comparison with other deep learning-based surrogates in terms of computational cost would further contextualize the trade-offs involved in adopting this approach.

Another limitation is the reliance on fully supervised learning, which requires a large number of high-fidelity training samples with ground-truth Cd values obtained from CFD or wind tunnel experiments.

While the study evaluates DragSolver across multiple datasets, it does not explicitly address real-world deployment challenges, such as robustness to noisy or incomplete 3D scans. Industrial CAD models and real-world scans often contain missing data, occlusions, or sensor artifacts that could impact the model’s predictions.

Hence, while this is a great concept, further research is needed to optimize computational efficiency, reduce data dependency, and improve robustness to real-world noise.

**Questions For Authors:**

None

**Relation To Broader Scientific Literature:**

The paper is focused on automotive drag prediction and as such narrow in scope.

**Theoretical Claims:**

There were no theoretical claims in the paper.

---

> ### Author Rebuttal · Authors · 2025-03-31
>
> We sincerely appreciate the reviewer’s insightful and constructive comments. Here, we respond to each concern in detail.
>
> **Note to reviewers:**  We conducted additional experiments and analyses in response to your valuable suggestions. These results will be explicitly included in the revised manuscript. Detailed experimental configurations are provided in [**Table 1(Click to View)**](https://anonymous.4open.science/r/ICMLRebuttal-F329/Table%201.jpg), and supplementary results can be found anonymously here:  [**Anonymous Supplementary Tables**](https://anonymous.4open.science/r/ICMLRebuttal-F329)
>
> ---
>
> **Q1: Computational Efficiency and Comparison with Deep Learning-based Surrogates**
>
> **A1:** We appreciate the reviewer’s comment on computational efficiency. The reported inference time (0.9–5 seconds per shape) includes 10 inference passes for uncertainty estimation and data loading overhead. To address this concern, we conducted additional efficiency comparisons with state-of-the-art models (PTV3, Mamba3D, PointGPT; [**Tables 2**](https://anonymous.4open.science/r/ICMLRebuttal-F329/Table%202.jpg) and [**3**](https://anonymous.4open.science/r/ICMLRebuttal-F329/Table%203.jpg)). DragSolver consistently achieves superior accuracy, particularly under limited training data (5%–30%) and high-noise conditions where other models often fail to converge. A single-pass inference (without uncertainty estimation) for the entire test set (1154 samples) takes only 11.98–13.25 seconds, demonstrating DragSolver’s practical computational efficiency. Additionally, automotive aerodynamic optimization typically requires feedback at intervals of seconds or minutes during design iterations rather than millisecond-level speed, thus DragSolver fully meets real-world requirements. We will clarify this context explicitly in the revised manuscript.
>
> **Q2: Data Dependency (Fully Supervised Learning)**
>
> **A2:** We appreciate the reviewer’s insightful comment regarding data dependency. Although our current method indeed uses fully supervised learning and thus relies on labeled data, we specifically evaluated DragSolver’s effectiveness under significantly reduced training data conditions (as low as 5%–30%, see [**Table 2**](https://anonymous.4open.science/r/ICMLRebuttal-F329/Table%202.jpg)). The results demonstrate DragSolver’s remarkable ability to maintain high accuracy (*R²*>0.92 at 10% training data) and superior stability compared to state-of-the-art surrogate models (PTV3, Mamba3D, and PointGPT), which largely fail to converge at these limited training ratios. These findings highlight DragSolver’s efficiency in leveraging limited labeled data. Nevertheless, integrating semi-supervised or physics-informed learning approaches to further reduce data dependency remains an important future direction.
>
> **Q3: Robustness to Noise and Incomplete Data**
>
> **A3:** We appreciate the reviewer’s valuable suggestion regarding robustness to real-world noisy or incomplete 3D data. To address this important concern, we conducted extensive experiments with varying levels of realistic noise and data augmentation ([**Table 3**](https://anonymous.4open.science/r/ICMLRebuttal-F329/Table%203.jpg)), including substantial random dropout (up to 40%), rotations (±4°), translations (0.03), and noise addition (9%) to simulate common real-world imperfections such as missing data, occlusions, or sensor artifacts. The results clearly show that DragSolver consistently achieves superior accuracy and stability under these challenging conditions, significantly outperforming state-of-the-art surrogate models (PTV3, Mamba3D, and PointGPT), which often fail to converge effectively under higher noise intensities. This strongly supports DragSolver’s robustness and suitability for real-world deployment. Nevertheless, explicitly testing DragSolver on industrial CAD data and actual 3D scans remains an essential direction for future research, which we will highlight in the revised manuscript.
>
> **Q4: The Paper is Narrow in Scope (Automotive Drag Prediction)**
>
> **A4:** We appreciate the reviewer’s valuable point. Automotive drag prediction itself is an important and challenging problem with significant industrial impact, directly influencing vehicle energy efficiency and emissions. Moreover, automotive aerodynamics is widely recognized as a classical and representative scenario in aerodynamic shape optimization. Although our current study specifically targets automotive applications, the methodologies proposed in DragSolver—such as multi-scale feature extraction, surface-guided gating, and uncertainty quantification—can naturally generalize to broader aerodynamic and hydrodynamic design problems (e.g., aerospace, naval, and structural engineering). We will clarify both the intrinsic importance of automotive drag prediction and the potential broader applicability of our methods in the revised manuscript.

---

### Official Review · Reviewer_bnLA · 2025-03-14

**Overall Recommendation:** 4

**Summary:**

In this paper, the authors propose DragSolver—a method to effectively estimate physical properties of shapes, such as cars, without the need to run expensive CFD simulations, allowing for the design of novel shapes much faster and more effectively. The proposed method consists of four major blocks: (1) multi-scale feature extraction, which enables more accurate predictions across multiple scales, (2) input normalization, allowing the model to work with various meshes of changing size and number of nodes/edges, (3) an effective approach to ignoring irrelevant parts of the shape—such as interior objects—to produce more precise predictions, and (4) an MC-Dropout method that enables the approach to estimate uncertainties, which are of great importance in industrial workflows. The method is compared against a number of modern 3D architectures and demonstrates superior performance in terms of physical properties prediction accuracy.

**Claims And Evidence:**

The authors make several claims about each major component of their method (mentioned 4 parts) and either adequately discuss the motivation behind these claims and/or support them experimentally later in the experimental section.

**Essential References Not Discussed:**

No essential references are missing.

**Experimental Designs Or Analyses:**

The design of the experiments sufficiently covers different aspects of the proposed approach, including both the accuracy of in-distribution predictions and the generalization ability of the model. Rigorous ablations are also much appreciated and support the claims made about the method. Additionally, the method is compared against a number of popular 3D architectures, further strengthening the case for its effectiveness.

**Methods And Evaluation Criteria:**

The choice of datasets and benchmarks is adequate and accurately reflects the current state of research in this direction. The evaluations are rigorous and demonstrate the effectiveness of the approach from multiple perspectives.

**Other Comments Or Suggestions:**

N/A

**Other Strengths And Weaknesses:**

In short, the major Strengths of the paper are:

* The proposed DragSolver framework is clearly structured, with each component thoroughly justified and experimentally supported. The paper includes rigorous ablation studies that strengthen its claims.

* The experiments cover multiple aspects, including in-distribution accuracy and generalization ability, and the method is compared against modern 3D architectures, demonstrating its effectiveness.

* The paper thoroughly discusses prior work, highlights the limitations of existing approaches, and evaluates the method on relevant, up-to-date datasets, ensuring its alignment with the current state of research.

**Questions For Authors:**

N/A

**Relation To Broader Scientific Literature:**

The paper clearly situates itself within the existing literature by thoroughly discussing prior work on estimating physical properties of shapes, particularly in the context of CFD-free predictions. It provides sufficient detail on previous approaches, highlighting their limitations and demonstrating how the proposed **DragSolver** framework addresses these challenges, namely through the four discussed components of the method.

**Theoretical Claims:**

No theoretical claims, theorems, or proofs are presented in the paper.

---

> ### Author Rebuttal · Authors · 2025-03-31
>
> We sincerely appreciate the reviewer’s positive evaluation and constructive summary, which greatly encourages us.
>
> **Note to reviewers:**  We conducted additional experiments and analyses in response to the valuable suggestions from other reviewers. Detailed experimental configurations are provided in [**Table 1(Click to View)**](https://anonymous.4open.science/r/ICMLRebuttal-F329/Table%201.jpg), and supplementary results can be found anonymously here:  [**Anonymous Supplementary Tables**](https://anonymous.4open.science/r/ICMLRebuttal-F329)
>
> Based on insightful feedback from other reviewers, we have further strengthened our manuscript with the following detailed analyses and clarifications:
>
> - **Computational Efficiency:** We explicitly compared DragSolver against state-of-the-art deep learning surrogate models (PTV3, Mamba3D, PointGPT) under limited training data and varying noise conditions. These comparisons clearly demonstrate DragSolver’s superior predictive accuracy and competitive inference speed ([**Table 2**](https://anonymous.4open.science/r/ICMLRebuttal-F329/Table%202.jpg) and [**Table 3**](https://anonymous.4open.science/r/ICMLRebuttal-F329/Table%203.jpg)).
>
> - **Robustness under Limited Data and Noise:** We evaluated DragSolver extensively under significantly reduced training samples (as low as 5%–30%) and challenging noise scenarios (up to 40% dropout, ±4° rotation, 9% random noise), confirming its consistent robustness and strong generalization capabilities ([**Table 2**](https://anonymous.4open.science/r/ICMLRebuttal-F329/Table%202.jpg) and [**Table 3**](https://anonymous.4open.science/r/ICMLRebuttal-F329/Table%203.jpg)).
>
> - **Uncertainty Quantification:** We clarified and differentiated epistemic (MC Dropout) from aleatoric (random sampling) uncertainties through additional targeted experiments, emphasizing their complementary roles in robust uncertainty estimation ([**Table 6**](https://anonymous.4open.science/r/ICMLRebuttal-F329/Table%206.jpg)).
>
> - **Choice of Point Cloud Representation:** We provided explicit rationale and empirical evidence supporting our choice of point clouds over meshes, highlighting advantages in computational efficiency, flexibility, and generalizability for large-scale automotive aerodynamic analysis ([**Table 4**](https://anonymous.4open.science/r/ICMLRebuttal-F329/Table%204.jpg)).
>
> - **Wheelbase Normalization:** We further clarified the practical necessity and benefits of our wheelbase normalization strategy through comparative experiments, confirming it significantly reduces training instability due to scale variations while preserving original geometric proportions ([**Table 5**](https://anonymous.4open.science/r/ICMLRebuttal-F329/Table%205.jpg)).
>
> We believe these comprehensive revisions effectively address key concerns, further improving the manuscript’s clarity, rigor, and broader impact.
>
> Again, we sincerely thank the reviewer for their encouraging and supportive comments.

---

> > ### Comment · Reviewer_bnLA · 2025-04-03
> >
> > Thank you for your thoughtful response to the reviews. I appreciate the clarifications provided and the additional experiments. After reading your rebuttal as well as the comments from the other reviewers, I continue to believe that this work could be an interesting contribution. In particular, the area of surrogate modeling for CFD lacks broadly adopted baselines, and I find it valuable that your method also produces uncertainty estimates. I therefore maintain my original score.

---

> > > ### Author Response · Authors · 2025-04-06
> > >
> > > Thank you sincerely for your thorough and supportive review in the first round. Your positive evaluation and recognition of the potential impact of our work have been greatly encouraging for us, and we truly appreciate your thoughtful insights and constructive feedback.
> > >
> > > Currently, we find ourselves facing a difficult situation: as the reviewer discussion period is approaching its end and the acknowledgment deadline (April 4, AoE) has already passed, we have not yet received further responses from the other two reviewers. Given this circumstance and considering ICML’s reviewing policy, we now only have this opportunity to directly communicate with you and potentially seek your further support.
> > >
> > > During the rebuttal stage, we invested significant effort and computational resources—conducting rigorous supplementary experiments on eight A100 GPUs running continuously for several days—to fully address all reviewer comments. Specifically, we provided extensive additional comparisons with state-of-the-art methods (e.g., PTV3, Mamba3D, PointGPT), robust analyses under limited data (down to 5%) and significant noise conditions (up to 40% dropout, ±4° rotation, and 9% random noise), and offered deeper clarifications regarding our uncertainty quantification approach. As a result, we believe the manuscript is now considerably stronger, and the experimental validations are more comprehensive.
> > >
> > > Given these substantial enhancements and additional efforts, could we kindly ask whether you feel the manuscript has improved sufficiently in rigor and significance to justify increasing your score? We would deeply appreciate your consideration and additional support.
> > >
> > > Thank you once again for your valuable time and effort.

---

### Decision · Program_Chairs · 2025-05-01

**Decision:**

Accept (poster)

**Comment:**

The paper introduces a transformer-based method for estimating the automotive drag coefficient from 3D vehicle models as an alternative to traditional CFD simulations.
Reviewers initially had concerns about the use of point clouds over meshes and the lack of comparison to more traditional CFD-based approaches, rather than the provided comparisons to other learning-based approaches.
However, the paper provides substantial benchmark experiments where DragSolver demonstrates good performance (in and out-of-distribution) compared to other recently proposed learning-based approaches (point cloud / mesh-based, transformer or SSM backbone).
The authors provide rigorous ablations and added additional ones during the rebuttal phase, alleviating many of the raised concerns by the reviewers.
Even though this paper is an edge case, I think it is a valuable contribution and I urge the authors to release the code upon acceptance as promised.